# Discrepancies in Persistent Dry Eye Signs and Symptoms in Bilateral Pseudophakic Patients

**DOI:** 10.3390/jcm8020211

**Published:** 2019-02-07

**Authors:** Akiko Hanyuda, Masahiko Ayaki, Kazuo Tsubota, Kazuno Negishi

**Affiliations:** 1Department of Ophthalmology, Keio University School of Medicine, Tokyo 160-8582, Japan; tsubota@z3.keio.jp; 2Epidemiology and Prevention Group, Center for Public Health Sciences, National Cancer Center, Tokyo 104-0045, Japan; 3Department of Nutrition, Harvard T.H. Chan School of Public Health, Boston, MA 02215, USA; 4Otake Clinic Moon View Eye Center, Kanagawa 242-0001, Japan

**Keywords:** dry eye disease, cataract surgery, ocular surface distress, maximum blinking interval, tear instability

## Abstract

Despite the increased awareness of early prophylaxis and treatment for dry eye disease (DED) during the first few weeks after cataract surgery, the chronic effect of cataract surgery on the risk of ocular surface abnormalities has not been fully explored. This study was to assess the prevalence of DE subjective symptoms and clinical tests according to the cataract surgery. A total of 172 patients who underwent bilateral cataract surgeries at least 5 months before the recruitment date and 1225 controls with no cataracts were evaluated for their subjective DE symptoms (dry sensation, foreign-body sensation, ocular pain, ocular fatigue, sensitivity to bright light, and blurred vision) and ophthalmic parameters (tear break-up time, keratoconjunctival staining scores, and maximum blinking interval). The presence of subjective DE symptoms was generally inversely associated with cataract surgeries, whereas abnormal clinical tests were more pronounced among postsurgical cataract patients than among controls. Pseudophakic patients showed a 57% increased prevalence of severe keratoconjunctivitis, compared to controls (*P* = 0.02). In contrast, among subjective DE symptoms, significantly lower odds of sensitivity to bright light were detected among cases than controls; the multivariable-adjusted odds ratio (95% confidence interval) comparing pseudophakic patients with noncataract patients was 0.56 (0.34–0.92) (*P* = 0.02). In conclusion, persistent tear instability and corneal epitheliopathy were found even at several months or more after cataract surgery. This study demonstrates the importance of evaluating ocular surface conditions in pseudophakic patients, even if they lack DE symptoms.

## 1. Introduction

Dry eye disease (DED) is a multifactorial disease of the tears and ocular surface that affects patients’ ocular and general health, well-being, and quality of life [1]. The prevalence of DED ranges from approximately 5% to 50% worldwide and has been shown to be dramatically higher in people aged 50 years old or older [2]. Given the considerable growth in the global aging population, there is an increased awareness of the increased risk of DED after cataract surgery [3,4,5,6,7,8,9,10], which is one of the most frequently performed surgeries in elderly people [11]. It has been reported that the incidence of DED after phacoemulsification is as high as 10% [5] or even 30% [10] among patients who were free of DED preoperatively. One study found that over 60% of patients who underwent cataract surgery had a short tear break-up time (TBUT), while 76% demonstrated corneal epitheliopathy [12]. Another study showed a marked shortage of goblet cells and microvilli in damaged conjunctival epithelia after cataract surgery [4]. Miyake and his group also reported a significant increase in corneal staining scores and a decrease in TBUT at one month postoperatively [10].

In general, dry eye signs and symptoms may be detectable immediately after surgery and can result from surgical invasion, such as injured corneal nerves, acute inflammation, ocular surface distortion, microscopic light, or local anesthetics [3,4,5,6,7,8,9,10]. These perioperative dry eye symptoms gradually decreased over approximately 1–2 months with topical application of postoperative medications, including steroids and nonsteroidal anti-inflammatory drugs (NSAIDs) [10], and as a result of the natural healing process [3,10,13,14,15,16]. However, conversely to improvement in dry eye subjective symptoms, we observed that a certain population of pseudophakic patients demonstrated persistent dry eye signs even at several years after cataract surgery. Thus, we speculated that long-lasting ocular surface problems might be associated with cataract surgery.

Although some patients may be dissatisfied with their postoperative result due to a suboptimal refractive outcome encompassing ocular surface discomforts, a growing body of evidence suggests that numerous positive effects accompany cataract surgery [17,18,19]. In addition to the marked increase in visual acuity, it is widely accepted that cataract surgery may improve a patient’s cognitive as well as motor functions [19], increase the quality of their sleep [18], and even alleviate depression [17].

Given that surgical intervention may induce and worsen DED, early detection and adequate treatment for ocular surface abnormalities are critical to improve the patient’s perceived surgical outcome. Nonetheless, obtaining a clinical determination of postoperative ocular surface dysfunction can be challenging because the signs and symptoms of dry eye can often be discordant [20]. Furthermore, most previous studies that have assessed the association between cataract surgery and subsequent DED had follow-up times of 3 months postoperatively [3,10], during which the tissue healing process may be ongoing; thus, the validity of these results for chronic outcomes of surgical interventions is unconvincing [21]. Hence, this study was performed to investigate the association between the long-term effects of cataract surgery and the prevalence of ocular surface disorders among bilateral pseudophakic patients at a time point at least 5 months after cataract surgery and to compare this population with noncataract subjects. In exploratory analyses, we evaluated whether subjective dry eye symptoms and objective ocular surface abnormalities, including tear instability, blinking interval, and keratoconjunctival staining scores, differ according to the presence of cataract surgical history.

## 2. Materials and Methods

### 2.1. Study Design, Ethical Approval, and Study Population

This study was a multisite, hospital-based, cross-sectional case-control study conducted from April 2015 to May 2016. The cases and control subjects were recruited from five clinical sites in Komoro Kosei General Hospital (Nagano, Japan), Shinseikai Toyama Hospital (Toyama, Japan), Tsukuba Central Hospital (Ibaraki, Japan), Jiyugaoka Ekimae Eye Clinic (Tokyo, Japan), Todoroki Eye Clinic (Tokyo, Japan), and Takahashi-Hisashi Eye Clinic (Akita, Japan).

The respective institutional review boards and ethics committees of Shinseikai Toyama Hospital (Permit Number: 150503) and Komoro Kosei General Hospital (Permit Number: 2705) approved this study, and this study was conducted in accordance with the tenets of the 1995 Declaration of Helsinki (as revised in Edinburgh, 2000). Informed consent was obtained from all participants.

### 2.2. Inclusion and Exclusion Criteria

All cases and controls were aged over 50 years old and had a best-corrected visual acuity of at least 20/25 bilaterally. Controls were selected from each eye clinic. We excluded participants who had incomplete data on subjective or objective dry eye symptoms and signs, those aged over 85 years old, those with a treatment history of DED beyond using artificial tears or any history of ocular diseases or ocular surgeries except uncomplicated cataract surgeries, and those with any ocular medications except hyaluronic acid, diquafosol, rebamipide, or steroids at baseline. Patients who had any major intraoperative/postoperative complications or who underwent a cataract surgery within five months were also excluded from the cases to minimize the direct effect of invasion of surgery.

### 2.3. Experimental Protocol

We extracted information from the participants’ medical records regarding basic demographic features and clinical treatment information. In all cases, an experienced surgeon performed a standard phacoemulsification and aspiration with implantation of a posterior chamber intraocular lens (IOL) under topical anesthesia. A 2.75 mm clear corneal or corneoscleral incision and a side port approximately 1 mm in size, located 90 degrees away from the main incision, were made. A foldable IOL was implanted in the capsular bag. All patients were treated with the same postoperative regimen of topical steroids (0.1% betamethasone), NSAIDs (0.1% bromfenac or 0.1% diclofenac), and antibiotics (levofloxacin or moxifloxacin) for the first four postoperative weeks and only NSAIDs for the subsequent two months (only a few cases with prolonged inflammation were continued the use of NSAIDs up to five months postoperatively).

### 2.4. Ophthalmological Examinations and Interviews of Subjective Symptoms

In each clinic, ophthalmic parameters, including best-corrected visual acuity, six subjective symptoms (dry sensation, foreign-body sensation, ocular pain, ocular fatigue, sensitivity to bright light, and blurred vision), and three objective signs, including the mean TBUT, keratoconjunctival staining scores (0–9 points) based on the Japanese dry eye diagnostic criteria [22], and maximum blinking interval (MBI) [23]. TBUT was evaluated twice, and the mean value was determined. Corneal and conjunctival fluorescein staining scores were evaluated in three areas (the temporal bulbar conjunctiva, nasal bulbar conjunctiva, and cornea) and scored on a 0–3-point scale in each section (0: no damage to 3: damaged entirely); the scores were summed up to a maximum of 9 points in total [22]. MBI was defined as the blinking interval when we asked the patients to keep their eyes open [23]. Our previous studies and other investigations have suggested that a short MBI was related to the severity of dry eye status [23,24,25]. Ocular surface abnormality was defined as a TBUT ≤5 s, keratoconjunctival staining scores ≥3 points, and MBI <10 s, based on the Japanese dry eye diagnostic criteria [22] as well as previous studies [1].

### 2.5. Statistical Analysis

We used unpaired *t*-tests for the univariable analyses of continuous demographic and clinical features. For univariable categorical variables, Chi-squared tests were used to compare the frequencies according to the history of bilateral cataract surgeries.

To assess the association of dry eye symptoms and clinical features with the history of bilateral cataract surgeries, univariable and two multivariable adjusted logistic regression analyses were performed to obtain odds ratios (ORs) and 95% confidence intervals (CIs). Considering the significant difference in age distribution between no cataract surgery and postcataract surgery groups, we adjusted for age (in the 10-year age group) and sex (men vs. women) in the first model (model 1). In the second model, we additionally adjusted for any use of dry eye medications (yes vs. no) (model 2). For each dry eye symptom or ophthalmic parameter, we evaluated associations with any symptomatology or abnormal test with a reference group that had no particular dry eye symptom or clinical finding.

To further evaluate the association between an individual’s dry eye status and their cataract surgical histories, Spearman correlation analyses were conducted. In the exploratory analyses, we restricted subjects who lacked subjective dry eye symptoms and examined the association between MBI and other dry eye objective signs in the strata of bilateral cataract surgical histories. All statistical tests were two-sided, and the significance level was set to an α of 0.05. All analyses were performed using SAS software (Version 9.4, SAS Institute, Cary, NC, USA).

## 3. Results

We enrolled 172 patients who underwent bilateral cataract surgeries and 1225 subjects who had no cataract in either eye as the cases and controls, respectively. The mean postoperative duration was 2.1 ± 3.4 years (0.5–21 years) in the cases. The demographics and clinical features of the cases and controls are shown in Table 1. The mean age was significantly older in cases than in controls. In general, subjective dry eye symptoms were less prevalent among cases. In contrast, cases presented significantly higher keratoconjunctival staining scores but shorter TBUT and MBI than were found in the controls. The frequencies of abnormal ocular surface parameters in the controls and cases were 11% and 16.5% (*P* = 0.008), respectively, for corneal epitheliopathy; 11% and 13.6% (*P* = 0.13), respectively, for short TBUT; and 11.2% and 14.9% (*P* = 0.05), respectively, for abnormal MBI.

Table 2 shows the univariable and multivariable logistic regression analyses of subjective and objective dry eye symptoms and signs in relation to cataract surgical histories. After controlling for age, sex, and dry eye medications, the individuals who underwent bilateral cataract surgeries were less susceptible to subjective dry eye symptoms but more likely to present worse clinical dry eye signs than was observed in the controls. Compared to controls, pseudophakic patients showed a 57% increased prevalence of severe keratoconjunctivitis (*P* = 0.02). Among subjective dry eye symptoms, sensitivity to bright light was remarkably different between cases and controls; the multivariable-adjusted OR (95% CI) comparing pseudophakic patients with noncataract patients was 0.56 (0.34–0.92) (*P* = 0.02). These discrepant findings were generally consistent between the sexes, although the statistical power was limited to reaching a significant level for most variables (Appendix A).

In addition, Spearman correlation coefficients indicated that there was a significant correlation between subjective symptoms and objective parameters among controls but inconsistent among cases (Table 3).

Considering that previous studies suggested that the important role of the involuntary compensatory reflex mechanism was associated with DED [26], we further assessed the association between MBI and other ocular dry eye parameters among subjects who lacked subjective dry eye symptoms (Appendix A). Notably, regardless of the history of cataract surgery, there were statistically significant correlations between MBI and both TBUT and ketatoconjunctival staining scores among individuals who lacked each of the subjective dry eye symptoms. 

## 4. Discussion

In this hospital-based observational study, we investigated the association between ocular surface dysfunction and the chronic effect of postcataract surgery. Compared with noncataract surgical subjects, postoperative cataract patients had significantly greater keratoconjunctival staining scores and shorter TBUT and MBI. In addition, the present study suggested that there is a distinct difference between dry eye symptoms and clinical test results for ocular surface abnormalities in bilateral pseudophakic patients, even after a relatively long postoperative period. Although subjective symptoms and objective signs were generally discordant among postoperative patients, we found that chronic corneal epitheliopathy might be associated with certain subjective symptoms, including dry sensation, foreign-body sensation, and ocular pain. Our current results suggest that the long-term effects of cataract surgery show a greater prevalence of ocular surface dysfunction, which is likely to be undiagnosed due to the minimal symptoms.

Although it is increasingly recognized that there are differences in findings among dry eye signs and symptoms within the few weeks after cataract surgery [10,27], little is known about whether the influence of cataract surgical invasion persistently affects ocular surface conditions, in disagreement with subjective symptoms. One possible explanation for why fewer dry eye subjective symptoms were observed among the pseudophakic patients in this study may be that corneal sensitivity was decreased due to advanced age and neurodegeneration following surgical intervention. Previous studies suggested that the severity of postoperative inflammation, the duration of suffering DED, and aging might share a complex interrelationship with corneal sensitivity [27,28,29,30,31,32], which can induce subjective dry eye symptoms. In our present study, the mean age was approximately 10 years older in the cataract surgery group than in the no surgery group. These findings were consistent with the previous study that older age was a significant predictor for dry eye discordance with less subjective symptoms than objective signs [32]. Another plausible cause of the lack of postoperative subjective symptoms might be that the prolonged use of NSAIDs continued for up to five months after cataract surgery in a few cases. Hence, subjective symptoms, especially nonvisual symptoms, such as ocular pain, foreign-body sensation, and dry sensation, might have been relieved by NSAIDs treatment among postoperative cataract patients. In addition, some visual symptoms, including sensitivity to bright light and blurred vision, could be attributable to the cataract itself; thus, patients report improved subjective symptoms after successful cataract extraction.

Cataract patients may present a wide range of health problems, including visual disturbance, photophobia, sleep disorders, and psychological distress [17,18,19]. It has been reported that individuals with depression demonstrate more ocular dryness than objective indicators for DED [17], partly because psychological factors might significantly contribute to dry eye symptoms. One recent study showed that people who expressed greater subjective happiness were less likely to recognize DED even when they had abnormal ocular surface findings [33]. Furthermore, pseudophakic patients had a lower prevalence of ocular pain and depression [17,18]. Thus, it could be plausible that patients who underwent cataract surgery might be unaware of ocular surface discomfort because of the positive outcome of cataract surgery, in accordance with our current findings. Although additional studies are warranted, ophthalmologists should perform ongoing ocular surface examinations even after the acute phase of cataract surgery has passed because patients’ satisfaction with increased visual acuity might hinder the identification of ocular surface abnormalities over the long term.

Consistent with a recent report [34], significant correlations were found between MBI and other ocular surface parameters among patients who lacked dry eye subjective symptoms, independent of their history of cataract surgeries. One possible explanation is that a decreased blinking interval may potentially reflect an involuntary compensatory reflex associated with dry eye [26], which may increase friction and contribute a worsening DED, especially in postsurgical patients. In fact, we observed that MBI was significantly shorter in pseudophakic patients than in phakic subjects, a finding that was supported by previous results showing that the blink rate was higher in dry eyes than in normal eyes [23,24,26]. In addition, it is increasingly recognized that friction-related diseases (FRDs), such as superior limbic keratoconjunctivitis, lid wiper epitheliopathy, and conjunctivochalasis, are a major dynamic cause of DED [35,36]. A recent study suggested that there is a strong negative association between TBUT and FRD, regardless of the DED status or subtype [36]. Together with a previous finding showing that an increased frequency of blinking may be a possible driver for ocular surface distress [23,24,26], it is conceivable that pseudophakic patients who lack subjective symptoms might also lack early prophylaxis and treatment for dry eyes, which exacerbate their ocular surface parameters compared to those observed in individuals with prominent dry eye symptoms.

Meibomian gland dysfunction (MGD) has been suggested as another possible etiology for DED after cataract surgery [37,38]. The association between MGD and postoperative DED may include age-related changes in the Meibomian glands [39] as well as poor hygiene around the orbit after eye surgery. It is likely that many patients avoid cleaning the eyelids because they are afraid of unintentionally compressing the operated eye, and this might increase the prevalence of MGD after cataract surgery. Unfortunately, we did not evaluate the status of the Meibomian glands pre- and postoperatively in our study; however, a previous study also supported the notion that susceptibility to MGD is higher in older patients who underwent cataract surgery, and that it could lead to postoperative DED [40].

Our study has several limitations. First, because this is an observational study, unmeasured or residual confounding factors may remain. Nonetheless, our multivariable adjusted analyses of possible risk factors may attenuate potential errors. Second, due to the cross-sectional design of this study, we were unable to assess the causal relationship between observed dry eye discrepant findings and the history of cataract surgery. Furthermore, there is a fundamental limitation in the lack of data for corneal sensitivity because we failed to confirm whether fewer subjective dry eye symptoms were directly attributed to the persistent corneal nerve damages following the cataract surgical invasion or other relevant dry eye discordant factors such as advanced age, positive psychological effects, or use of postsurgical medications in the postcataract surgery group. Therefore, future longitudinal studies with pre- and postoperative ophthalmic evaluations, including corneal sensitivity tests, would be indispensable for gaining understanding of dry eye pathophysiology after cataract surgery. Third, because the majority of our participants were Japanese, our data may lack generalizability. Hence, additional studies performed in patients with different ethnicities are warranted to investigate the association between the chronic influence of cataract surgery on ocular surface distress that is characterized both subjectively and objectively. In addition, we have acknowledged the significant age difference between individuals who underwent bilateral cataract surgeries and those with their own lenses. Thus, we have conducted age-/sex-adjusted and multivariable adjusted models in addition to univariable models and the results were interpreted in a cautious manner. Finally, potential selection bias and heterogeneity among intergroups might not be completely eliminated, although the distribution of baseline characteristics except age was not substantially different between cases and controls.

This study has several strengths. First, our detailed data on dry eye subjective symptoms and clinical manifestations were all evaluated by a single experienced dry eye specialist (M.A.) according to the most frequently used and standardized Japanese dry eye criteria, and this may have maximized its internal and external validity. Second, the samples were collected from multiple institutions in Japan, allowing us to conduct a large-scale case-control study including enriched ophthalmic parameters in a rigorous manner. The novelty of our current study is that we eliminate the direct influence of cataract surgery and subsequent acute inflammatory reactions and successfully capture the long-term characteristics of dry eye symptoms and signs by excluding individuals who underwent cataract surgeries within five months.

In conclusion, we suggest that cataract surgery has a persistent harmful effect on tear stability and ocular surface, which are prone to being masked by a favorable outcome following cataract extraction. Our findings suggest that excessive blinking and a shorter MBI might be good proxies for estimating ocular surface distress even though the patients failed to demonstrate existing dry eye symptoms. Adding a careful perioperative examination, early intervention, and adequate management for postoperative DED could play a key role in protecting the ocular surface and improving long-term patient satisfaction with cataract surgery.

## Figures and Tables

**Table 1 jcm-08-00211-t001:** Baseline demographic and clinical features of the study participants.

Characteristics	All Participants	No Cataract Surgery	Postcataract Surgeries	*P* Value ^†^
Participant, *n* (%)	1397	1225	172	
Mean age in years (SD)	66.6 (9.7)	65.4 (9.1)	75.4 (9.0)	<0.001
Sex, *n* (%)				0.12
Men	531 (38.0)	475 (38.8)	56 (32.6)	
Women	866 (62.0)	750 (61.2)	116 (67.4)	
Eye drop users for dry eye symptoms, *n* (%)	411 (29.4)	359 (29.3)	52 (30.2)	0.80
Subjective symptoms				
Dry sensation, *n* (%)	336 (24.1)	302 (24.7)	34 (19.8)	0.16
Foreign-body sensation, *n* (%)	263 (18.8)	226 (18.5)	37 (21.5)	0.34
Ocular pain, *n* (%)	84 (6.0)	75 (6.1)	9 (5.2)	0.65
Ocular fatigue, *n* (%)	445 (31.9)	396 (32.3)	49 (28.5)	0.31
Sensitivity to bright light, *n* (%)	238 (17.0)	217 (17.7)	21 (12.2)	0.07
Blurred vision, *n* (%)	258 (18.5)	227 (18.5)	31 (18.0)	0.87
Clinical features				
Tear break-up time, seconds (SD)	4.6 (1.8)	4.7 (1.8)	4.3 (1.9)	0.02
Keratoconjunctival staining score ^‡^, (SD)	0.31 (0.6)	0.29 (0.6)	0.44 (0.7)	0.008
Maximum blinking interval, sec (SD)	12.0 (5.8)	12.1 (6.0)	10.9 (4.4)	0.002
Abnormal ocular surface parameters *				
Short tear break-up time, *n* (%)	172 (12.3)	75 (11.0)	97 (13.6)	0.13
Corneal epitheliopathy, *n* (%)	172 (12.3)	117 (11.0)	55 (16.5)	0.008
Short maximum blinking interval, *n* (%)	172 (12.3)	107 (11.2)	65 (14.9)	0.05

^†^ The tests for significance were t-tests for continuous variables and Chi-square tests for categorical variables. ^‡^ Keratoconjunctival staining scores, which originally ranged from 0 (minimum) to 9 (maximum) points by the Japan Dry Eye Association, were categorized into 3 groups: “mild” for 0–2 points, “moderate” for 3 points, and “severe” for 4+ points. We then rescored patients with “mild”, “moderate”, and “severe” as 0, 1, and 2 points, respectively. * Abnormal ocular surface parameters were defined as follows: tear break-up time ≤5 s, the presence of corneal epitheliopathy (keratoconjunctival staining score ≥3 points according to the Japanese dry eye criteria), and maximum blinking interval <10 s. Abbreviations: SD, standard deviation.

**Table 2 jcm-08-00211-t002:** Univariable and multivariable logistic regression analyses of dry eye subjective symptoms and objective signs in relation to the history of cataract surgeries.

	All Participants					
Characteristics	Univariable OR (95% CI) *	*P* Value	Age, Sex-Adjusted OR (95% CI) *	*P* Value	Multivariable OR (95% CI) **	*P* Value
**Subjective symptoms**						
Dry sensation	0.75 (0.51–1.12)	0.16	0.90 (0.59–1.37)	0.61	0.96 (0.63–1.48)	0.86
Foreign-body sensation	1.21 (0.82–1.79)	0.34	1.05 (0.69–1.59)	0.83	1.06 (0.70–1.61)	0.79
Ocular pain	0.85 (0.42–1.72)	0.65	0.88 (0.42–1.84)	0.72	0.89 (0.42–1.89)	0.77
Ocular fatigue	0.83 (0.59–1.19)	0.31	0.95 (0.66–1.38)	0.79	0.96 (0.66–1.39)	0.82
Sensitivity to bright light	0.65 (0.40–1.04)	0.07	0.56 (0.34–0.92)	0.02	0.56 (0.34–0.92)	0.02
Blurred vision	0.97 (0.64–1.46)	0.87	0.82 (0.53–1.26)	0.36	0.82 (0.53–1.26)	0.36
**Objective signs** ^†^						
Short tear break-up time	1.28 (0.93–1.76)	0.13	1.24 (0.87–1.75)	0.23	1.24 (0.88–1.76)	0.22
Corneal epitheliopathy	1.59 (1.13–2.26)	0.008	1.53 (1.05–2.21)	0.03	1.57 (1.08–2.30)	0.02
Short maximum blinking interval	1.39 (1.00–1.94)	0.05	1.06 (0.75–1.51)	0.73	1.07 (0.75–1.52)	0.71

^†^ Abnormal ocular surface parameters were defined as follows: tear break-up time ≤5 s, the presence of corneal epitheliopathy (keratoconjunctival staining score ≥3 points according to the Japanese dry eye criteria), and maximum blinking interval <10 s. * Adjusted for age (in 10-year age groups), sex (male vs. female). ** Adjusted for age (in 10-year age groups), sex (male vs. female), and the use of dry eye medications (yes vs. no). Abbreviations: CI: confidence interval; OR: odds ratio.

**Table 3 jcm-08-00211-t003:** Association between dry eye subjective symptoms and objective signs according to bilateral cataract surgical histories *.

	Tear Break-Up Time		Keratoconjunctival Staining Score		Maximum Blinking Interval	
Characteristics	Coefficient	*P* Value	Coefficient	*P* Value	Coefficient	*P* Value
**No Cataract Surgery**						
Dry sensation	−0.20	<0.001	0.21	<0.001	−0.10	<0.001
Foreign-body sensation	−0.14	<0.001	0.17	<0.001	−0.08	0.004
Ocular pain	−0.13	<0.001	0.14	<0.001	−0.11	<0.001
Ocular fatigue	−0.12	<0.001	0.17	0.009	−0.09	0.002
Sensitivity to bright light	−0.09	0.001	0.07	0.01	−0.09	0.001
Blurred vision	−0.09	0.002	0.07	<0.001	−0.06	0.03
**Postcataract Surgeries Bilaterally**						
Dry sensation	−0.12	0.10	0.19	0.01	−0.11	0.16
Foreign-body sensation	−0.05	0.55	0.27	<0.001	−0.11	0.14
Ocular pain	−0.003	0.97	0.26	<0.001	−0.07	0.39
Ocular fatigue	−0.09	0.23	−0.03	0.69	−0.13	0.09
Sensitivity to bright light	−0.12	0.12	0.04	0.57	−0.02	0.82
Blurred vision	−0.14	0.06	0.11	0.14	−0.008	0.92

* Spearman correlation coefficients between subjective symptoms and objective signs.

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
