# Peer review of "Discrepancies in Persistent Dry Eye Signs and Symptoms in Bilateral Pseudophakic Patients"

_jcm, 2019, doi:10.3390/jcm8020211_

Round 1
Reviewer 1 Report
Reviewer finds this an interesting and well-written report.
There is one possible concern in view of the significant difference in age between "No Cataract Surgery" and "Postcataract Surgery" subjects, the Authors might wish to discuss the assumptions and mathematical methods used to adjust for Age in compiling Table 2.
Author Response
Response to Reviewer 1 Comments
Thank you very much for your interest in our manuscript entitled “Discrepancies in Persistent Dry Eye Signs and Symptoms in Bilateral Pseudophakic Patients”. To aid in the re-review of this manuscript, we have included a point-by-point response to each comment. The reviewer’s comments are italicized and placed in square brackets. In addition, within the revised manuscript, we have used underlined text to highlight changes in response to the reviewers’ comments.
We appreciate the suggestions and comments by the reviewer. As a consequence of valuable suggestions, we believe that our manuscript has been much improved.
[Reviewer 1: Reviewer finds this an interesting and well-written report.]
[Point 1. There is one possible concern in view of the significant difference in age between "No Cataract Surgery" and "Postcataract Surgery" subjects, the Authors might wish to discuss the assumptions and mathematical methods used to adjust for Age in compiling Table 2.]
We appreciate the reviewer’s comments on our intriguing study and new insights for the discordant dry eye subjective and objective signs and symptoms among pseudophakic patients. We believe that our study can provide further insights on multifactorial features of dry eye pathophysiology in a relatively long-term after cataract surgery. In response to the reviewer’s comment, we have modified the MATERIALS AND METHODS and DISCUSSION sections as follows:
“Considering the significant difference in age distribution between no cataract surgery and post-cataract surgery groups, we adjusted for age (in the ten-year age group) and sex (men vs. women) in the first model (model 1). In the second model, we additionally adjusted for any use of dry eye medications (yes vs. no) (model 2).”
(MATERIALS AND METHODS, page 7)
“In our present study, the mean age was approximately 10 years older in cataract surgery group than no surgery group. These findings were consistent with the previous study that older age was a significant predictor for dry eye discordance with less subjective symptoms than objective signs.[32]”
(DISCUSSION, pages 9–10)
“In addition, we have acknowledged the significant age difference between individuals who underwent bilateral cataract surgeries and those with their own lenses. Thus, we have conducted age-/sex- adjusted and multivariable adjusted models in addition to univariable model and the results were interpreted in a cautious manner. Finally, potential selection bias and heterogeneity among intergroups might not be completely eliminated, although the distribution of baseline characteristics except age was not substantially different between cases and controls.”
(DISCUSSION, page 12)

Reviewer 2 Report
Although the article addresses an interesting topic, various methodological limitations prevent to obtain clear data.
Abstract: Please add numerical data in the abstract.
Full text: The lack of corneal sensitivity values is a major limitation of the study, and this should be clearly stated in the paper. Lower values of sensitivity among post-cataract patients secondary to surgical nerve damage could justify why paeudophakic patients reported less discomfort symptoms compared to controls. Furthermore, it is not clear why patients used NSAIDS in the postoperative course of cataract surgery for a very long period (until 6 months). To my knowledge, this protocol does not follow current guidelines.
There is no information about the pre-operative clinical characteristics of patients who undergone cataract surgery. Were they “normal” or did they experienced DED signs and/or signs already before surgery?
Were pseudophakic patients using tear substitutes during the study period?
The difference of age between the two groups is a crucial limitation of the study, and this should be clearly stated in the paper. The author themselves highlighted the strong relationship between age and DED in the introduction section!!!
The lower frequency of DED symptoms in pseudophakic patients compared to controls is rather curious for me. Usually in the clinical practice cataract surgeons are not able to satisfy patients’ expectations for the opposite reason (good visual acuity but DED symptoms after uneventful cataract surgery).
Author Response
Response to Reviewer 2 Comments
Thank you very much for your interest in our manuscript entitled “Discrepancies in Persistent Dry Eye Signs and Symptoms in Bilateral Pseudophakic Patients”. To aid in the re-review of this manuscript, we have included a point-by-point response to each comment. The reviewer’s comments are italicized and placed in square brackets. In addition, within the revised manuscript, we have used underlined text to highlight changes in response to the reviewers’ comments.
We appreciate the suggestions and comments by the reviewer. As a consequence of valuable suggestions, we believe that our manuscript has been much improved.
[Reviewer 2: Although the article addresses an interesting topic, various methodological limitations prevent to obtain clear data.]
[Point 1. Abstract: Please add numerical data in the abstract.]
We appreciate the comment. In response to the reviewer’s comment, we have modified the ABSTRACT as follows:
“Pseudophakic patients showed a 57% increased prevalence of severe keratoconjunctivitis, compared to controls (P=0.02). In contrast, among subjective DE symptoms, significant lower odds of sensitivity to bright light were detected among cases than controls; the multivariable-adjusted odds ratio (95% confidence interval) comparing pseudophakic patients with noncataract patients was 0.56 (0.34–0.92) (P=0.02).”
(ABSTRACT, page 2)
[Point 2. Full text: The lack of corneal sensitivity values is a major limitation of the study, and this should be clearly stated in the paper. Lower values of sensitivity among post-cataract patients secondary to surgical nerve damage could justify why paeudophakic patients reported less discomfort symptoms compared to controls.]
We appreciate the reviewer’s comment. We recognize the limitation of the lack of corneal sensitivity data in our current study. In response to the reviewer’s comment, we have modified the DISCUSSION section as follows:
“Furthermore, there is a fundamental limitation in the lack of data for corneal sensitivity because we failed to confirm whether fewer subjective dry eye symptoms were directly attributed to the persistent corneal nerve damages following the cataract surgical invasion or other relevant dry eye discordant factors such as advanced age, positive psychological effects, or use of postsurgical medications in the post-cataract surgery group. Therefore, future longitudinal studies with pre- and post- operative ophthalmic evaluations including corneal sensitivity tests would be indispensable for gaining understanding of dry eye pathophysiology after cataract surgery.”
(DISCUSSION, pages 11–12)
[Point 3. Full text: Furthermore, it is not clear why patients used NSAIDS in the postoperative course of cataract surgery for a very long period (until 6 months). To my knowledge, this protocol does not follow current guidelines.]
We appreciate the comment. According to the current guideline (Lim BX, et al. Cochrane Database Syst Rev. 2016), most of the cases in our present study were treated with nonsteroidal anti-inflammatory drugs (NSAIDs) for approximately up until 3 months following the cataract surgery. With the expertise in the postoperative use of NSAIDs, one of the authors (M.A.) prescribed NSAIDs for about 4–5 months only in limited cases owing to the prolonged post-surgical inflammation (Italian Diclofenac Study Group. J Cataract Refract Surg. 1997). Due to the cross-sectional study design, we acknowledge the limited information on post-operative treatment history. In response to the reviewer’s comment, we have modified the MATERIALS AND METHODS and DISCUSSION sections as follows:
“All patients were treated with the same postoperative regimen of topical steroids (0.1% betamethasone), NSAIDs (0.1% bromfenac or 0.1% diclofenac), and antibiotics (levofloxacin or moxifloxacin) for the first 4 postoperative weeks and only NSAIDs for the subsequent 2 months (only a few cases with prolonged inflammation were continued the use of NSAIDs up to 5 months postoperatively).”
(MATERIALS AND METHODS, page 6)
“Another plausible cause of the lack of postoperative subjective symptoms might be that the prolonged use of NSAIDs continued for up to 5 months after cataract surgery in a few cases.”
(DISCUSSION, page 10)
[Point 4 Full text: There is no information about the pre-operative clinical characteristics of patients who undergone cataract surgery. Were they “normal” or did they experienced DED signs and/or signs already before surgery?]
We appreciate the comment. Given the nature of cross-sectional study design, we lacked detailed information on pre-operative ocular surface status among cases. Hence, future longitudinal studies with pre- and post-operative ophthalmic evaluations should be warranted to assess the causal link between persistent dry eye signs and the history of cataract surgery. Nonetheless, we excluded those who had severe dry eye disease (DED) beyond using artificial tears at the recruitment date and the baseline characteristics between cases and controls were not substantially different (Table 1). In response to the reviewer’s comment, we have modified the DISCUSSION section as follows:
“Second, due to the cross-sectional design of this study, we were unable to assess the causal relationship between observed dry eye discrepant findings and the history of cataract surgery. Furthermore, there is a fundamental limitation in the lack of data for corneal sensitivity because we failed to confirm whether fewer subjective dry eye symptoms were directly attributed to the persistent corneal nerve damages following the cataract surgical invasion or other relevant dry eye discordant factors such as advanced age, positive psychological effects, or use of postsurgical medications in the post-cataract surgery group. Therefore, future longitudinal studies with pre- and post- operative ophthalmic evaluations including corneal sensitivity tests would be indispensable for gaining understanding of dry eye pathophysiology after cataract surgery.”
(DISCUSSION, pages 11–12)
[Point 5. Full text: Were pseudophakic patients using tear substitutes during the study period?]
We appreciate the comment. In our current study, 30.2% of pseudophakic patients used dry eye medications including tear substitutes as shown in Table 1.
[Point 6 Full text: The difference of age between the two groups is a crucial limitation of the study, and this should be clearly stated in the paper. The author themselves highlighted the strong relationship between age and DED in the introduction section!!!]
We appreciate the comment. In response to the reviewer’s comment, we have modified the DISCUSSION section as follows:
“In our present study, the mean age was approximately 10 years older in cataract surgery group than no surgery group. These findings were consistent with the previous study that older age was a significant predictor for dry eye discordance with less subjective symptoms than objective signs.[32]”
(DISCUSSION, pages 9–10)
“In addition, we have acknowledged the significant age difference between individuals who underwent bilateral cataract surgeries and those with their own lenses. Thus, we have conducted age-/sex- adjusted and multivariable adjusted models in addition to univariable model and the results were interpreted in a cautious manner. Finally, potential selection bias and heterogeneity among intergroups might not be completely eliminated, although the distribution of baseline characteristics except age was not substantially different between cases and controls.”
(DISCUSSION, page 12)
[Point 7. Full text: The lower frequency of DED symptoms in pseudophakic patients compared to controls is rather curious for me. Usually in the clinical practice cataract surgeons are not able to satisfy patients’ expectations for the opposite reason (good visual acuity but DED symptoms after uneventful cataract surgery).]
We appreciate the comment. As the reviewer pointed out, we have recognized that some of patients had severe dry eye symptoms after cataract surgery. In general, such symptomatic dry eye patients were more detectable compared with asymptomatic patients in a relatively short period after cataract surgery and thus clinicians appeared to prevent exacerbating their DED in an early stage of the disease cause. Nonetheless, in our clinical experience, we also found a number of pseudophakic patients who demonstrated tear instability and keratoconjunctivitis with no ocular discomforts during the check-ups for 6 month or more after cataract surgery. It is increasingly recognized that cataract surgery might significantly improve not only vision-related quality of life but also cognitive, mental, and physiological functions in the older people (Ishii K, et al. Am J Ophthalmol. 2008; Ayaki M, et al. Rejuvenation Res. 2015). With the emerging evidence suggesting that dry eye subjective symptoms were inversely associated with mental health (Vehof J, et al. Ophthalmology 2017; Kawashima M, et al. PloS One 2015), we have speculated that cataract surgery might affect differently dry eye signs and symptoms in a relatively long-term. Hence, we have conducted this study to examine the susceptibilities of dry eye subjective symptoms and objective signs among bilateral pseudophakic patients at a point at least 5 months after cataract surgery. We believe that our present study can inform future studies to elucidate the chronic effects of cataract surgical intervention on the ocular surface pathophysiology.

Round 2
Reviewer 2 Report
I congratulate the Authors for their detailed responses and for the clear changes done in the revised manuscript. However, I have only a last comment... Since the paper is mainly focused on discomfort symptoms, why did the Authors investigated 6 subjective parameters rather than to use a conventional validated questionnaire? For example, the Speed questionnaire that is not influenced by vision task...
Author Response
Hanyuda A et al. JCM-424308R
Response to Reviewer 2 Comments
Thank you very much for your interest in our manuscript entitled “Discrepancies in Persistent Dry Eye Signs and Symptoms in Bilateral Pseudophakic Patients”. To aid in the re-review of this manuscript, we have included a point-by-point response to each comment. The reviewer’s comments are italicized and placed in square brackets. In addition, within the revised manuscript, we have used underlined text to highlight changes in response to the reviewers’ comments.
We appreciate the suggestions and comments by the reviewer.
[Reviewer 2: I congratulate the Authors for their detailed responses and for the clear changes done in the revised manuscript.]
[Point 1. However, I have only a last comment... Since the paper is mainly focused on discomfort symptoms, why did the Authors investigated 6 subjective parameters rather than to use a conventional validated questionnaire? For example, the Speed questionnaire that is not influenced by vision task....]
We appreciate the comment on our intriguing study and new insights for the discordant dry eye subjective and objective signs and symptoms among pseudophakic patients. As we referred in our “MATERIALS AND METHODS” section (page 6), the six subjective symptoms used in our current study were selected from the most frequent dry eye specific symptoms in the Dry Eye Questionnaire Score (DEQS), which was previously developed based on the latest definition and diagnostic criteria of the Japanese Dry Eye Society in 2006 (Shimazaki J et al. J. Eye. 2007) and validated with high specificity and reproducibility (Sakane Y et al. JAMA Ophthalmol. 2013). In addition, these six subjective symptoms were most common symptoms among dry eye disease (DED) patients in our outpatient clinic (the Department of Ophthalmology, Keio University Hospital, Tokyo) and were well documented in a number of studies in a Japanese population (Kaido M et al. Invest Ophthalmol Vis Sci. 2016; Miyake K et al. Clin Ophthalmol. 2017; Ayaki M et al. Int J Ophthalmol. 2018). Given the nature of this clinic-based observational study, there is an unmet physician’s need that our findings are not only comparable to the current Japanese diagnostic criteria but applicable to a daily practice. Although future studies are warranted to support our results by using the whole set of the validated questionnaire, the primary purpose of this study was to evaluate the findings from our clinical practice and generalize the importance of DED management for all cataract surgeons, regardless of their subspecialty of DED. Accordingly, we believe that our study can provide further insights on dry eye pathophysiology in a relatively long-term after cataract surgery.
